# Personalization and Localization in Human-Robot Interaction: A Review of Technical Methods

**Mehdi Hellou** [1,2], **Norina Gasteiger** [1,3], **Jong Yoon Lim** [1], **Minsu Jang** [4] **and Ho Seok Ahn** [1,*]

1   Department of Electrical, Computer and Software Engineering, The University of Auckland, Auckland 1010, New Zealand; helloumehdi@outlook.fr (M.H.); n.gasteiger@auckland.ac.nz (N.G.); jy.lim@auckland.ac.nz (J.Y.L.)
2   School of Engineering, The University of Manchester, Manchester M13 9PL, UK
3   School of Health Sciences, The University of Manchester, Manchester M13 9PL, UK
4   Electronics and Telecommunications Research Institute (ETRI), Daejeon 34129, Korea; minsu@etri.re.kr
*   Correspondence: hs.ahn@auckland.ac.nz; Tel.: +64-9923-7860

**Abstract:** Personalization and localization are important when developing social robots for different sectors, including education, industry, healthcare or restaurants. This allows for an adjustment of robot behaviors according to the needs, preferences or personality of an individual when referring to personalization or to the social conventions or the culture of a country when referring to localization. However, there are different models that enable personalization and localization presented in the current literature, each with their advantages and drawbacks. This work aims to help researchers in the field of social robotics by reviewing and analyzing different papers in this domain. We specifically focus our review by exploring different robots that employ distinct models for the adaptation of the robot to its environment. Additionally, we study an array of methods used to adapt the nonverbal and verbal skills of social robots, including state-of-the-art techniques in artificial intelligence.

**Keywords:** human-robot interaction; social robotics; personalization; localization; adaptation

## 1. Introduction

The study of human-robot interaction (HRI) focuses on the integration of autonomous systems in our lives and how humans can easily interact with robots by taking into account all aspects of the environment. This area considers several sectors in robotics, such as rehabilitation [1], exoskeletons [2] and collaborative robots [3,4]. Each sector deals with general issues that we may encounter when developing robotic services for humans. One of the main issues remains the safety of the users, considering how developers need to design their robots in order to accomplish their main tasks while remaining safe. Some examples include making the machine aware of human motions and gestures in order to not disrupt their activity or collide with them [3] or to support the user's work and contribute to their health, e.g., helping them to avoid lifting heavy objects [2,3] or manipulating dangerous products [4]. Safety is not the only matter when designing robots for human-robot collaboration or interaction. Robotic services also need intuitive interfaces and services to be useful. These could include a virtual simulation to visualize steps needed for the robot and the human operator to assemble a product [4] or an exoskeleton arm for upper-limb rehabilitation [1]. Either way, the device has to include an element of interactivity. This becomes increasingly important when the robot and the human work collaboratively or when social interactions between robots and humans are required.

Social robotics refers to the development of social robots that are able to help people in their daily lives and adapt their behaviors to each user's needs, preferences and personality. In order to have such technology, we need systems that have the ability to adapt to their environments and to the other actors within these environments. The purpose of this paper is therefore to review technical methods that could enable such technology. More

particularly, we divide our review into two aspects of social robots and the elements that could impact their behaviors: personalization and localization.

A personalized robot specializes its skills to a particular user or a set of users in order to provide necessary help. It can draw on different elements from the user, such as preferences, needs, personality, previous experiences, attitudes and/or habits so as to adopt appropriate behaviors according to the situation. Localization refers to the adaptation of a product to a country or region. It integrates the notion of "culture", defining a group of individuals from a country by the different social rules established between them, e.g., how people greet one another or how they prefer particular products to be designed. We characterize robots with these abilities as adaptive social robots (ASR). ASRs are defined as, "an autonomous or semiautonomous robot where speech is controlled by a human operator through a Wizard of Oz (WoZ) setup that can be termed as a decision-making engine capable of perceiving the user information from the environment" [5,6]. Thus, an adaptive robot is not utterly autonomous but can be semiautonomous by employing information gathered from humans to adapt and adjust its behaviors.

The above concepts encourage us to consider which techniques to use when integrating these abilities for a social robot. More precisely, we need to analyze the state-of-the-art-methods available in the literature so that we can improve social robots. Here, we will review a collection of technical papers that present methods to enable localization and personalization by summarizing various techniques and methodologies, from rules-based systems (RBS) to artificial intelligence (AI) methodologies. First, we present the different information used for personalization and localization. Specifically, we explore the different models employed by robots for adaptation. We also determine the different technical methods used for personalization and localization by distinguishing RBS and AI methods before general conclusions are made.

## 2. Methodology

To locate relevant papers, we electronically searched digital platforms such as Google Scholar and Microsoft Academic Scholar. Most of our chosen papers are from Google Scholar due to the vast number of papers retrieved by this database. We used keywords such as "social robots", "adaptive social robots", "personalization social robots", "localization social robots", "adaptation autonomous systems", "methods for long-term interaction" and "adaptation in social robotics". We found a total of 27 papers to present. These were determined by discussion and relevance to the topic.

We analyzed the papers by determining which type of factors each paper used for adaptation. Similar to the study in [7], we focused on adaptation by distinguishing between the user model and social model. Further, we characterized the methods as using static information or being dynamic. As mentioned in the introduction, we also analyzed the methods used for personalizing and localizing the robot's behavior by considering whether the technique was based on RBS or whether the robot employs more autonomous methods based on AI. An overview of the papers analyzed in this review is depicted in Table 1.

**Table 1.** Summary of the papers analyzed in this review.

| References | Robot | Model Used | Autonomy | Personalized/Adaptive Features |
|---|---|---|---|---|
| Gockley et al. [8] | Valerie Robot Receptionist | None | Semiautonomous | Speaks with users by using a personality and a preprogrammed background story. Gives information about the facilities. |
| Torrey et al. [9] | PEARL | Social model | Fully | Adapts questions and guides users to retrieve the right tools according to their cooking level. |
| Gross et al. [10] | TOOMAS | User and social model | Semiautonomous | Personalized to each shopper when looking for particular products (navigation and interactive communication). The speed of the robot is also personalized and depends on the speed of the person following. |
| Lee et al. [11] | Snackbot robot | None | Semiautonomous | Moves in an open environment. Delivers snacks and speaks to people. |
| Tan et al. [12] | Robot butler, specific to the project. | User and social model. | Semiautonomous. | Robot butler that can detect and track humans. Recognizes hand gestures, serves beverages and performs dialog conversation with guests. |
| Farahmand et al. [13] | N/S | None | Fully | Generation of correct behaviors according to the tasks and abilities of the robot. Develops a repertoire of behavior modules. Organizes behavior modules in the agent's architecture. |
| Kanda et al. [14] | N/S | Social and user model | Semiautonomous | Provides guidance, entertainment and advertises services for users based on their preferences. Adapts behavior to be socially acceptable in Japan (e.g., always greeting by recalling the name of the user before starting the interaction). |
| Huang et al. [15] | N/S | Social model | Semiautonomous | Introduction of a new ROS (robotics operating systems) toolkit to generate social behaviors for the robot. |
| Lee et al. [16] | Snackbot robot | User model | Semiautonomous | Same features as in [11]. Additionally, the robot can remember and recall previous users' information. |
| Pieskä et al. [17] | CENTRIA robot | None | N/S | Uses a database for restaurant's menus and a server for ordering logic. Integration of a tablet with various applications to entertain customers. |
| Aly et al. [18] | NAO | Social model | Fully | Personality recognition. Adjusts nonverbal and verbal behaviors according to user's personality. |
| Giuliani et al. [19] and Keizer et al. [20] | Robot bartender | Social model | Fully | Serves beverages, recognizes engaged user, dialogue to take customer's drink. |
| McColl et al. [21] | Brian 2.1 | User and social model | Fully | Greets user by their name. Tracks meal consumption. Detects users' attention. Motivates the user to finish their meal. |
| Sekmen et al. [22] | Pioneer 3-AT mobile robot | User model | Fully | Proposes drinks to users according to their preferences and the time of the day. Reminds of meeting and class times. |

**Table 1.** *Cont.*

| References | Robot | Model Used | Autonomy | Personalized/Adaptive Features |
|---|---|---|---|---|
| Shiomi et al. [23] | N/S | Social and user model | Semiautonomous | Coordinates the work and tasks of different types of robots for elderly care. |
| Andrist et al. [24] | NAO | Social model | Fully | Analyzes gaze aversion in HRI and adopts the appropriate behavior. |
| Portugal et al. [25] | Robot specific to the project | Social and user model | Fully | Different services provided to help quality of life of elderly people: facial recognition, emotion recognition, stores relevant information about the user (e.g., name, medicines). |
| Pieters et al. [26] | NAO | Social model | Fully | Predicts human intention. Adapts robot's behavior based on the human's task. |
| [27] | NICO, the Neuro-Inspired COmpanion | User model | Fully | Learns new objects with human help. Personalizes robot's behavior by remembering and recalling previous information concerning user (e.g., name, preferences). |
| Görür et al. [28] | N/S | Social model | Fully | Detects the intention of a human. Provides assistance according to the intention. |
| Perera et al. [29] | Pepper | User Model | Fully | Increases the awareness of the robot about its environment and enhances its ability to interact with people. Converses with them and recognizes familiar people. |
| Qureshi et al. [30] | Pepper | Social model | Fully | Learns how to greet people. |
| Liu et al. [31] | N/S | Social model | Fully | Learns both human-initiated and robot-initiated behavior for a social robot. Uses human-human example of shopkeeper interactions. |
| Tuyen et al. [32] | Pepper | Social model | Fully | Learns incrementally how to generate bodily gestures regarding individual users' cultural traits. Uses a human model to learn how to generate these gestures. |
| Yoon et al. [33] | NAO | Social model | Fully | Generates a sequence of gestures according to speech. |
| Foster et al. [34] | Pepper | Social model | Fully | Interacts with people in a shopping mall. Detects and tracks people. Detects people's social state (engagement). Responds to users' requests but also discusses open-domain topics. Offers route guidance and description. |
| Reig et al. [35] | Robot prototype | User model | Semiautonomous | Re-embodiment and co-embodiment configurations of different personalities inside the same robot. Adopts a personality according to the user's preferences. Uses different voices, images, and names based on the user's personality. |

### 3. Services for Personalization, Localization and Adaptation

As explained in Section 1, personalization and localization may refer to different elements used for adjusting the behavior of a robot. However, both of these terms need a fully adaptive system that is able to analyze its environment and adopt different strategies according to the situation. Thus, adaptation is required to facilitate personalization and localization.

*3.1. Adaptation for Social Robots*

We can find many definitions for "adaptation" that are not directly related to the domain of robotics. For example, in biology, adaptation is the ability of a living being to adjust to an environment by changing its physiology and behavior. The environment may be unknown and complex for the agent, which forces it to adapt to survive. We can transfer this definition to robots, but instead of surviving, robots need to achieve a task (or several). This may include a mobile robot that has to clean rooms or a robotic arm that has to lift objects and place them in a particular location.

In HRI, ASR has been described as a system that possesses the capacity to understand and show emotions, communicate with high-level dialogue, learn/adapt according to user responses, establish a social relationship, react according to different social situations and have varying social characteristics and roles [36]. All of these definitions are particularly true and depict a social robot as an entity that requires the ability to interpret people's needs and to respond accordingly. In order to achieve this, the robot has to use several elements from the human user to make decisions [6].

They may include elements such as the user's profile, emotions, personality and past interactions. The decision-making process could also be more general, and the robot may base its decisions on situations or scenarios by not accounting for the user. Martins et al. [7] defined adaptivity as "a system's ability to perform its function in different scenarios by automatically changing its operational parameters accordingly". This can refer to simple adaptations, e.g., a robot that controls its motion speed in relation to the person following it [10], or more complex adaptations, e.g., a robot tailoring its behaviors to the personality of the user [18]. Moreover, adaptation becomes more difficult when several agents interact with one another [13] since this requires agents to share information and synchronize their actions.

*3.2. User-Adaptiveness*

As discussed earlier, the purpose of ASR is to assist users and help them achieve a specific task. This could include physical tasks, such as lifting or delivering an object, or social tasks. In both cases, the user can play an important role during the task, as they may collaborate with the robot, or they can be key to the robot's purpose. In all cases, we employ the term "user-adaptiveness" when the users and information related to them plays an important role in the adaptation. Martins et al. [7] defined "user-adaptiveness" as a system that can deal with different scenarios that emerge from user-related data (e.g., the user's identity, preferences and expertise). This can be summarized as the ability of a robot to adjust its parameters related to users' information.

Examples of user-adaptiveness are evident in the literature. For example, Sekmen et al. [22] present a mobile robot that learns about and subsequently adapts to the behaviors and preferences of the people with whom it interacts. In another context, Gross et al. [10] present an autonomous robot that assists shoppers who are looking for a particular product. The system combines autonomous navigation and interactive communication to help users.

*3.3. Type of Services*

As explained earlier, for the purpose of localization and personalization, we need to base adaptation on validating information or, as expressed here, on an accurate model. In this paper, we distinguish two types of models used to adapt a robot's behavior: the social model and the user model.

1.  Social model: This broadly refers to systems that prioritize social skills and human interaction. These systems can identify and ultimately cater to differences across predetermined cultural norms and behaviors (e.g., greeting by bowing or knowing when it is appropriate to make eye contact). Similarly, they can respond appropriately across a group of people by demonstrating relevant social skills (e.g., verbally greeting someone or standing up when they arrive).
2.  User model: This model is characterized by individual preferences and priorities. Here, the system is often able to adapt its communication/interaction style, to cater to each individual user (rather than the group they belong to or affiliate with). The user's information can concern preferences, personality or needs.

Regardless of whether the robot's adaptation is built according to the user's information or factors pertaining to social acceptance, these elements can evolve throughout the interaction. We further distinguish these models as static or dynamic models. The only difference resides in the fact that dynamic models change their parameters over time when needed, while static models constantly maintain the same parameters.

### 3.3.1. Services without User or Social Models

It is not necessary to use one of the models described above to create an adaptive robot; the system can interact by using direct feedback from users during an interaction and not employ a specific model to achieve personalization and/or localization. For example, Farahmand et al. [13] present an artificial cognitive architecture for adaptive agents that can use sensors to behave in a complex and unknown environment. Without using any information about the user, the model was tested to operate with complex tasks, such as lifting an object with multiple agents. The purpose was mostly to present an architecture capable of generating the right behavior and planning an accurate sequence of behaviors to achieve a specific task.

Another example is presented by Lee et al. [11] whereby the authors developed a Snackbot robot, which is able to deliver snacks to users. The robot did not draw on the users' information but instead used a web page to deliver the correct product to the right person.

In the restaurant sector, Pieskä et al. [17] presented a mobile robot called CENTRIA. This social-service robot is also intended be used in the home and for healthcare services for older adults. The designed system is based on the Kompai platform, a specific software for interactive services. By using its knowledge about the dishes served in restaurants and the ordering logic, the robot proposes to help employees by indicating which tables are willing to be served and what their orders are. It also proposes additional services to the customers, such as a menu for ordering food or entertainment applications, like games.

Besides these examples, many rely on a social and/or a user model for adaptation. This is due to the central role of users in HRI, whereby user information is crucial to fostering adaptation, as shown next.

### 3.3.2. Services with Only the Social Model

As stated in the previous section, the social model refers to social skills and human behaviors such as engagement. These cues are not related to each user but are instead global and define a group of people. When applied, a robot could use different social information that is mostly related to cues that it can find and experiment with different users. This may include the level of engagement during a conversation [20] or intention according to specific tasks [26,28]. This type of information may be based on different criteria, like nonverbal cues, e.g., head position to detect the user's intention respecting a specific task [28] or verbal cues, such as speech, to predict a user's personality [18].

### Static Model

In a large European project, Foster et al. [34] presented an overview of the different methods and technologies used in the MuMMER project. This EU-funded project aimed to develop a social robot designed to interact naturally and flexibly with users in public spaces

(e.g., shopping malls). In this project, the robot, Pepper (https://www.softbankrobotics.com/emea/en/pepper, accessed on 4 October 2021), was used for the experiments and presented abilities that can be deployed in several interactions. These include conversations on an open topic or services, like route guidance and shopping-related information. Even though the system is fully autonomous, the adaptation of the robot is only based on the engagement of the user during the interaction.

Liu et al. [31] used a deeper meaning of this engagement by using proactive behavior to determine when a robot needs to start or continue a conversation when interacting with a human. In order to illustrate their model, the authors used the scenario of a robot shopkeeper in which the robot had to learn different skills from human demonstrations and apply these in real-life situations.

Torrey et al. [9] used a humanoid robot called PEARL to tackle the issue of how social robots might use adaptive dialogue to advise, instruct, guide, test or interview a varied group of individuals. They exemplified a robot chef that guides a participant in the task of matching the name of cooking tools with their corresponding pictures. Based on the users' cooking knowledge, the robot was able to provide information about the utensils. Thus, the guiding and helping of the robot were adapted to the knowledge level of participants by using a questionnaire administered before the experiments.

Dynamic Model

In contrast to static models, dynamic models can change their parameters in order to improve the adaptation of a robot. Görür et al. [28] presented a system that is able to recognize the intention of humans based on the different sensors of the robot. Particularly, the authors define an architecture that integrates the notion of theory of mind into a robot's decision-making to infer a human's intention and then adapt to it. They illustrated their system by setting up a scenario in which the robot takes the lead to help (or not help) a user to pick up an object. Their model can handle two conditions: the condition where the human's intention is estimated incorrectly, and the true intention may be unknown to the robot, and the condition where the human's intention is estimated correctly but they do not want the robot's assistance. By estimating the social state of the user, such as through different cues (e.g., facial expressions or speech), the system determines the expected action: whether the human needs help grasping an object or whether they do not require help.

Similarly, Pieters et al. [26] introduced an artificial cognitive architecture used to understand the complete state of a human (physical, intentional and emotional) and interact (using actions and goals) in a human-cognitive manner. The research also tackled the ability of an autonomous system to detect the human's intention when completing a task. In order to have a good understanding of its surroundings and the different objects present in the environment, the system uses a knowledge base divided into two components. These include (1) a declarative memory to represent the beliefs, relations and intentions of the world, of humans and objects and (2) a procedural memory to describe information about events and instances that have occurred.

In [20], a robot called JAMES plays the role of bartender. The project focuses on the development of a system capable of interacting with several users at the same time by using the scenario of a bartender. The robot possesses several input sensors, such as a vision system that tracks the location, facial expressions, gaze, behavior and body language of all people in the scene in real time, along with a linguistic processing system combining a speech recognizer with a natural language parser to create symbolic representations of the speech produced by all users. The output generators include a speech generator to talk with users during the interaction and controllers for the robot's head and arms. More importantly, the system is controlled by two main components, which permit the robot to recognize the user's social state and then choose the actions that should be generated by the robot. Before the interaction, the user's engagement is predicted by the system using supervised learning methods pretrained on an annotated human-robot corpus [37].

It is also important that a robot adapts to the user's personality, as determined by Aly et al. [18]. Indeed, the authors focused their work on the adaptation of the robot's behaviors based on the level of sociability of an individual. They specifically focused their work on how to improve the adaptability of a robot according to the user's personality traits. They distinguish two types: the extraverted, when an individual tends to be sociable, friendly, fun-loving and talkative; and the introverted, when an individual tends to be reserved, inhibited and quiet. When the robot estimates the group wherein the user belongs, it chooses the appropriate behavior by tailoring its speech and gestures. The authors extracted the majority of personality traits according to the criteria discussed in [38], described as the Big Five Framework: openness to experience, conscientiousness, extraversion, agreeableness and neuroticism. For example, the authors stated that extroverts use more positive emotion words and show more agreement and compliments than introverts.

In another context, Qureshi et al. [30] focus on the issue of making a robot learn social skills by themselves without having a supervised teacher. By introducing a Multimodal Deep Q-network, they assessed the ability of a robot to learn how to greet people by themselves in combining two phases: data generation and training on those data. From the literature, we can conclude that it is challenging to find dynamic social skills that can be used for adaptation. We notice that intention or engagement are the most commonly used criteria used for adaptation, except for in [30] and [18].

### 3.3.3. Services with Only the User Model

Unlike social cues, service robots in which adaptability adheres to a user model may use more information to change their behaviors. They will base their adjustment on more personal information that defines a user, such as preferences or habits. Since the information can significantly change between each user, systems with user models might need to employ external tools to gather information. For example, they may use surveys or questionnaires and store them in databases in case of future interactions with the same users. There are two distinct sub-models, within the user-model domain: those in which parameters remain the same from the beginning and do not change (static), and those in which the parameters change during the interactions and where the user model is not stationary (i.e., dynamic).

#### Static Model

Static models are evident within user models and require gathering of information about the users only once before starting the interaction. Surveys and questionnaires may be used to obtain this kind of information. However, they are not changed during the interaction, possibly resulting in repeatable behaviors if the robot interacts with the same user several times.

Sekmen et al. [22] introduced a mobile robot that was able to learn about the behaviors and preferences of the people with whom it interacted. Its role was to help users by proposing a drink and reminding them of their meeting or class time. The authors used questionnaires to gather participants' preferences about their preferred drinking time during the day, the type of drink and the time of their meetings. In order to demonstrate the efficiency of their system, the authors proposed an experiment based on short interactions with two systems: a nonadaptive one and an adaptive one. Most of the participants evaluated the performance of the adaptive robot as better than the nonadaptive robot.

In Churamani et al. [27], the paper tackles the question of whether a robot's adaptation to a conversation to accommodate the user's preferences allows for natural interactions. For that, they used the "object learning" scenario where the user teaches different objects to the robot using natural language. In addition to the main learning task, the robot has the ability to remember and recall personal information, such as the name of the user, if they have already interacted with the robot. With the help of several HRI-related surveys and questionnaires (e.g., GODSPEED or UTAUT [39]), the majority of the participants

appreciated the personalized conversations with a robot and perceived it to be more intelligent and likeable.

Dynamic Model

In addition to using initial information about the user, dynamic models are able to change this information or add new information according to the number of interactions they have with the same users. This is demonstrated in Reig et al. [35], whereby a robot is able to have multi-person interactions. Indeed, the authors put into practice a service robot with different agents' personalities co-embodied inside the system. The different personalities are used to enable the robot to interact with two different people at the same time. Based on this, the authors set up HRI experiments in three types of scenarios: a healthcare clinic, a department store and a scenario of restaurant recommendations. They also experimented with three types of adaptations, including the one proposed in the paper that consists of a robot that can host multiple personalized AI assistants that are accessed by the users in all aspects of their lives. The results show that users widely accepted the integration of this life agent capable of co-embodied personalities inside the same robot, and they particularly appreciated the fact that they could change the agent's personality as they wished.

Facial recognition is also a good indicator by which to model the user profile, such as in Perera et al. [29]. The authors set up and depicted different methods and techniques used to improve the autonomy of a humanoid robot by increasing its awareness of the environment. By describing some techniques for facial recognition and navigation, the authors depict an example with a Pepper robot, showing how to combine different methods to improve the social interaction and behaviors of social robots.

Following their paper [11], Lee et al. [16] extended their studies with Snackbot in the scenario of a robot delivering snacks for users. They improved the robot's behaviors by enabling it to record information about the users and recall them during other interactions. For example, the robot could determine the snack that each user ordered the most and the number of interactions it had had with them. As a result, the robot could create personalized sentences for each user, such as, "I missed you during my snack deliveries [$n$] times so far. I am glad to finally see you again", or "I was thinking about my first month here. I realized that I broke down and made mistakes [$n$] times in front of you. Sorry for that, and thank you for being patient with me". Although the robot's dialogue was scripted and controlled remotely by an operator, the personalized interaction was primarily accepted and preferred by participants, and it permitted better rapport, engagement and cooperation with the robot than non-personalized interactions.

Canal et al. [40] developed an assistive robot that helps users in three tasks: assisted feeding, shoe fitting and jacket dressing. The robot performed tasks in different manners for each user based on their preference. The preferences included the speed at which the robot executed the task and its verbosity, i.e., whether the robot should speak or not and the number of sentences it produces during the interaction, which consisted of informing the user before each task was performed. In this paper, the authors wanted to analyze if the users could guess which execution of the task used their chosen preferences and if they were satisfied when it was the case.

### 3.3.4. Services with User and Social Models

Some systems do not just employ a social model or a user model to perform adaptation but instead combine both. These systems are more complex to develop, but the combination of both models may strengthen a robot's performance. Indeed, they have the ability to improve their behaviors when interacting with humans by using indications related to the individual and more general social conventions.

### Static Model

Shiomi et al. [23] developed a network that coordinates the tasks of different service robots to help older adults. We can see the network as a coordinator that assigns activities to robots based on the needs of users and the information obtained about them at the beginning (e.g., names or walking ability).

In the same domain of robotic assistance, McColl et al. [21] investigated user engagement and compliance during meal-time interactions with a humanoid robot (Brian 2.1), along with overall acceptance and attitudes towards the robot. By using pre-coded behaviors and an RBS tree for choosing, the robot had the ability to detect attention and motivate a user to eat or drink the items on a meal tray while also promoting the social dimensions of eating. For example, the robot could determine when a user was distracted and encourage them to eat by orienting their attention toward the dishes.

Additionally, Gross et al. [10] presented the development of a social robot that helps users in a shopping mall. More specifically, the system assists shoppers when looking for particular products by employing appropriate navigation and interactive communication and adapting its speed depending on the speed of the person following.

Tan et al. [12] depicted their work about the development of a robotic butler. The robot is able to detect and track humans, converse with them about their interests and preferences and provide specific information about the facilities. The users could get the robot's attention by waving their hands. The guests were surprised by the performance of the robot in recognizing their request, and they found it acceptable to use hand gestures for getting the robot's attention. They were also surprised that the robotic butler was able to greet them by name and talk about their favorite actor, movie, sports team and player.

### Dynamic Model

In the same context and domain as [10], Kanda et al. [14] proposed a social robot in a shopping mall that is able to provide guiding services to users but also advertise different shops in the mall (based on users' preferences) and build rapport with them in a personalized interaction. Unlike [10], the robot could dynamically store preferences of the user, such as their preferred products or shops, and use them to advertise stores during the next interaction.

In the domain of healthcare for the elderly, Portugal et al. [25] presented an interactive mobile robot in the context of a collaborative European project called "SocialRobot". The project focused on providing practical and interactive robotic solutions to improve the quality of life of elderly people. The system was able to recognize the identity of and emotion expressed by a user and use personal information about users to adapt its behaviors. As explained in the paper, the robot was able to suggest calling a friend or caregiver according to a user's emotions. The authors plan to improve the capacity of the robot and conduct an experiment using real-world interactions.

As we can conclude by [14] and [25], dynamic models using social and user models require a substantial database to store any new information. These elements, fused with algorithms, would permit engineers to influence a robot's social skills, such as navigation, speech or gestures, whether they use nonverbal or verbal cues.

## 4. Methods for Adaptation

The papers presented in Section 3 may use different models to change a robot's behaviors; however, they share standard state-of-the-art methods well known in the current literature.

First, the designs of these methods are related to the autonomy of the systems, whether they are fully autonomous [20–22], semiautonomous [8,11] or controlled by a human operator [35]. This remains important for experimenters since it will influence the design of their methods and experiments, even if social impact and participants' reactions are their main interests.

Across all papers reviewed, we can distinguish two main components that could affect a robot's actions:

- Decision-making: relies on the robot's ability to make decisions based on the information provided by its sensors.
- Behavior generation: Rules that determine the behavior of the robot in social interaction. This includes the robot's gestures, speech, navigation or gaze, and is mainly used to enable the robot to perform effective humanlike communication.

The subsequent subsections present an overview of the different techniques that are commonly used.

### 4.1. Decision-Making

In order to have effective robots that can be deployed in the real world, they need to make decisions according to their environment. The environment may be unknown or known and often requires robots to use their sensors. Two distinct families relating to decision-making methods were evident in our review: rule-based and AI-based methods. Rule-based methods are related to human-crafted or curated rule sets, mostly known in the literature as rules-based systems (RBS). In contrast, AI-based methods use automatic rule inference to make decisions, such as machine learning. Both methods have various advantages and disadvantages.

#### 4.1.1. Rules-Based Systems

An RBS consists of specific rules to make decisions. They are often different from each other in practice while remaining similar in principle. Indeed, we have a potential view of RBS architecture in Figure 1. There are four main components in an RBS: a knowledge base (KB), an inference engine, a temporary working memory and a knowledge-acquisition module to add and subtract new information.

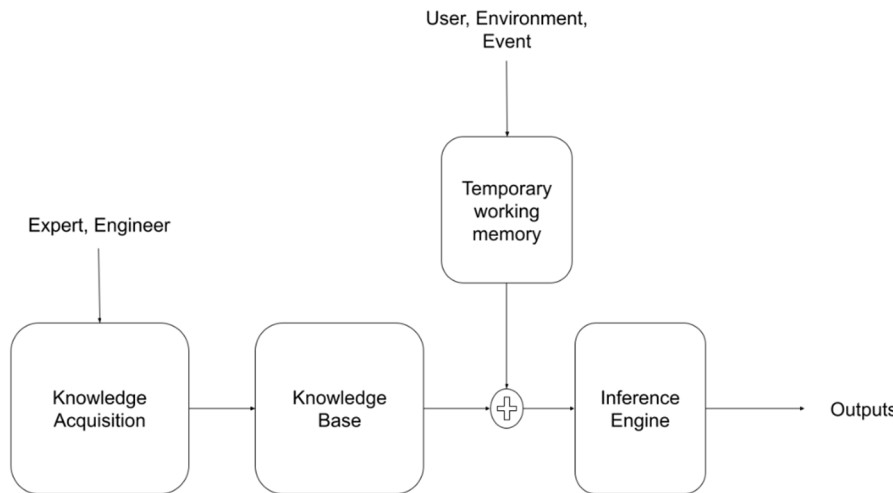

**Figure 1.** Overview of RBS architecture.

Knowledge Base

The knowledge base (KB) contains a set of rules and acts as the domain of knowledge for the system. These pieces of information are essential when developing adaptive social robots because they provide further information about the environments and the elements with which they will interact.

For example, the CENTRIA robot has a database containing the restaurant's menu, permitting the robot to propose different menu items to customers [17]. Databases can also use preprogrammed software, as specified in [9], that use the Artificial Intelligence Markup Language (AIML), an XML dialect for creating natural language software agents, in order to add knowledge of cooking tools, tool properties and answers to questions about tool properties. Knowledge may also employ blocks of codes that model a behavior's functioning [14] and how the robot's different modules interact to achieve specific behaviors. External tools and software can be used to describe the sequence of actions to accomplish a

particular behavior. There are several tools available to the public for research purposes in social robotics [15]. These tools are easily operable and can be changed to fit our needs, thus reinforcing their utility.

Inference Engine

An interference engine (which can be called a semantic reasoner) is responsible for interpreting rules, considering the current situation and taking action accordingly. Trees and graphs are most frequently used to model these rules and the links between them.

In [21], the authors investigated user engagement and compliance during meal-time interactions with a humanoid robot. The robot's decisions are modeled by using a rule-based tree that represents the different actions of the robot, such as to "encourage the user to drink".

RBS can be used to represent rules in chatbots, such as in [8], where a receptionist robot employs a rule-based pattern-matching system modified from Aine (http://www.neodave.civ.pl/aine/, accessed on 4 October 2021), which is, in turn, derived from AIML and ALICE (http://www.alicebot.org/, accessed on 4 October 2021). According to the authors, the rules are simple to write, can return any desired data (including tags usable by other components) and allow many different wordings of sentences to be recognized with just a few rules.

In [14], the authors investigate the employment of an RBS to model the behavior of a robot. For example, when letting their robot greet people, they used different condition-code blocks ("if" and "while") to determine the specific situations where the robot needs to greet a person. This inference method is efficient and quick to execute since the programmers already know the results of this algorithm and can fix it if there is an issue. However, the person needs to program all of the different behaviors and the conditions that enable the behaviors. Other autonomous AI methods are more efficient and do not require all conditions to be preprogrammed.

Similarly, a bartender robot in Giuliani et al. [19] used a conditional planner that works with incomplete information and sensing actions- the PKS (Planning with Knowledge and Sensing) (also mentioned in [41,42]). More specifically, PKS defines a robot's actions by a set of preconditions, which define the conditions that must be true for an action to be applied and capture the set of effects that the actions make when changing the robot's state. For example, when the robot needs to ask for a drink, PKS could use simple sentences, such as "ask-drink (customer)". Finally, PKS can use these predefined actions to construct plans by reasoning about actions using forward-chaining and creating logical plans utilizing the robot's knowledge. The particularity of PKS is based on the fact that the agent's knowledge (rather than the state of the world) is represented by a set of databases, and actions are represented as updates to these databases. This permits the modeling of actions as knowledge-level modifications to the agent's knowledge state rather than as physical-level updates to the world state.

Temporary Working Memory

Temporary working memory can also be integrated into an RBS, whereby the interference engine executes a production-system program. This component is where partial information acquired by the robot can be used to generate a behavior or complete a task during an interaction. In [15], the paper establishes a system that uses a "working memory", which stores data that other subsystems needed for processing. These data might refer to a context or an event to describe the current situation needed by other components to generate the appropriate behavior output.

Knowledge Acquisition

When a situation is repeatedly encountered by a system, the use of a long-term memory might be helpful to store these pieces of knowledge. Different operations, such as adding, subtracting or changing how input and output signals are received and sent, would

be possible with this type of memory. For example, in [14], a precoded episodic-memory module allows the robot to record different customers' information. This dialogue history permits the robot to have records about its interactions and recall shared memories with users (e.g., preferences).

Additionally, in [26], the authors used episodic memory to describe information about events and instances that occurred (e.g., what, where and when an event happened). This information is important for the system to use when confronted with the same situations.

Consequently, RBS is an efficient architecture whose results can be easily controlled by researchers, although these methods may be limited in terms of development and autonomy.

### 4.1.2. Artificial Intelligence Methods

In the previous section, we saw that the use of RBS enables social robots to make autonomous decisions without human assistance. However, a limit regarding their independence exists and requires preparing complex structures compared to the number of actions and tasks we want a robot to achieve. There are other learning methods, such as machine learning (ML), that can increase a robot's autonomy and achieve similar or even better results than RBS, even though their outputs cannot always be explained.

#### Machine Learning Methods

In the current literature, we distinguish different learning methods. Whether speaking of ML or deep learning (DL), these types of AI methods employ algorithms, building a mathematical model based on sample data in order to make predictions or decisions without being explicitly programmed to do so. In ML, we differentiate supervised learning and unsupervised learning. Supervised learning refers to models using labeled data to perform learning, which we can compare to a teacher showing a student the solution to solve a problem. On the other hand, unsupervised learning describes models availing a dataset with no pre-existing labels and minimal human supervision during learning.

For example, in [18], the authors employ a support vector machine (SVM) [43] to predict the user's personality based on their utterances. SVM is a model that uses a classification algorithm for a two-group classification problem, though it might be employed for multiclassification. The algorithm's objective is to find a hyperplane in an $n$—dimensional space ($n$—the number of features) that distinctly classifies the data points. Generally, the algorithm will attempt to determine the optimal parameters of the equation, defining the hyperplane by using a method of optimization to minimize a loss function, such as the "least squares method". In order to optimally minimize the loss of function, the algorithm needs a large set of data with their label. In this paper, SVM training consists of several corpora that map users' utterances to their personalities. For a more specific example, work presented in [38] focuses on the written style of different people covering the five personality traits described in the Big Five Framework (openness to experience, conscientiousness, extraversion, agreeableness, and neuroticism). This corpus enables the authors to state general relationships and characteristics of the five personality traits, e.g., conscientious people tend to avoid negations, negative emotion words and words reflecting discrepancies (i.e., should and would).

Using a more complex model, Liu et al. [31] present a neural network (NN) with an attention mechanism [44] to learn proactive behaviors for a social robot from human-human example interactions. They employ the system in a shopkeeper scenario, where the robot avails social skills, learning with the NN to sell cameras. In order to achieve this, the authors categorized the continuous stream of sensor data into typical behavior patterns to illustrate robot action and a joint-state vector for the NN. Subsequently, these joint-state vectors (three, to be exact) are provided as input sequences to the NN, which has the same structure as a multilayer perception neural network (MLPNN). As mentioned earlier, an attention mechanism is introduced to outline the joint-state vector (among the three most recent) that is the most relevant for the NN to predict the robot's action.

When data are not labeled and we do not explicitly know how to differentiate them, an NN can employ unsupervised learning algorithms. This was the case for Tuyen et al. [32], who analyzed different ways to make a robot learn in an unsupervised manner. Their robot learned different gestures for generating behaviors by using humans as an example. In doing so, they used the self-organizing map (SOM) [45], a technique to cluster a set of inputs corresponding to bodily gestures based on their similarities for expressing emotions. To allow long-term HRI and enable the robot to learn new gestures without corrupting the existing model incrementally, the authors used an alternative method for the training phase, called dynamic cell structure (DCS) neural architecture [46]. When the robot is accurately trained to learn these emotional gestures from the human model, they are then transformed to fit a robot model to improve its behaviors. To assess their model, the authors set up two types of experiments, whereby the first assessed the robot's ability to imitate similar gestures from the human model. During the experiment, the robot captured different motions produced by participants and attempted to imitate these gestures. The second experiment was divided into two trials, wherein during the first phase, the participants had to interact with Pepper. During the interaction, the robot determined the user's emotions through facial expressions and at the same time, recorded the participant's gestures. The second phase consisted of generating different emotional gestures (sad and happy), which were assessed in an online survey.

Reinforcement Learning Methods

Another branch of ML called reinforcement learning (RL) permits an agent to take actions in an environment to maximize the notion of cumulative reward. The agent learns to achieve a goal in an uncertain, potentially complex environment. The environment is typically stated in the form of a Markov decision process (MDP) because many RL algorithms for this context utilize dynamic programming techniques [47]. New methods have been introduced, such as Q-learning (QL) [48] and adaptive heuristic critic (AHC) algorithms [49], employed in many autonomous and multiagent applications. In social robotics, RL methods may be useful when the robot interacts in an environment dominated by uncertainty and performs actions with little human supervision. For example, Keizer et al. [20] use different MDPs to model the actions of a bartender robot whose role is to serve drinks to users. They employ two MDPs: for single-user cases and multiuser cases. Both MDPs have their own state spaces, each defined by a set of state features determined by the robot's sensors and their own action sets, corresponding to the range of decisions that the robot can make at the two stages, with two policies mapping states to actions. For instance, the policy used in single-user cases includes actions to interact with one user (e.g., greeting the user or serving a drink). In contrast, in the multiuser cases, actions are appropriate for a group of individuals, such as processing the user's order or telling the customer to wait. For the sake of facilitating the learning for both policies, the authors included a joint reward that is computed for each user served by the robot and summed at the end. The reward function takes into consideration whether the robot was successful (or not) in serving a user, the time taken to start the interaction with an engaged user and to achieve the task, as well as social penalties to acknowledge different discrepancies during a complete interaction. These may include situations such as when the system turns its attention to another user/customer while already speaking to another one. As explained earlier, the authors employed the premise of the QL method by encoding the policies as functions that associate a value to each state–action pair, called Q-values. Q-values are estimated by using cumulative rewards from the reward function. The optimal policies are found by using a Monte Carlo control algorithm [50].

Similarly, in [28], the authors draw on a partially observable Markov decision processes (POMDP) to define a robot's decision-making based on a human's intentions. As mentioned in Section 3.3.2, a POMDP includes states, such as a human's beliefs concerning a plan through a goal, e.g., "needs help in finding the object," actions representing both human

and robot actions and rewards that are introduced in the form of human emotional reactions to the robot (i.e., approval or disapproval).

Combining RL methods and DL models with NN can also illustrate a social robot's action, as presented in [30]. More specifically, the robot learns how to greet a person by using a multimodal deep Q-network (MDQN). It includes a dual-stream convolutional neural network (CNN) to approximate the action-state values through the robot's cameras to learn the optimal policy with QL. The dual stream obtained from the robot's camera enables the CNN to process the gray scale and the depth information. The robot can execute four legal actions regarding the action set, i.e., waiting, looking towards humans, waving its hand and handshaking with a human. In doing so, the reward function evaluates the success of the robot when the handshaking event occurs. More specifically, the function proposed by the authors gives a reward of 1 on the successful handshake, $-0.1$ on an unsuccessful handshake and 0 for the rest of the three actions. Ultimately, the authors implemented the QL method to make sure that the robot learned the optimal policy.

Other Methods

Other methods have been integrated into social robots to develop their social skills by combining the ones presented above or using other probabilistic methods, such as Bayesian networks (BN) or evolutionary theories.

In [13], the authors investigate the employment of an artificial cognitive architecture for adaptive agents that can use sensors to behave in a complex and unknown environment. The framework is a hybridization of reinforcement learning, cooperative coevolution, and a culturally inspired memetic algorithm for the automatic development of behavior-based agents. The authors introduce two different parts to separate the problem: (1) developing a repertoire of behavior modules and (2) organizing them in the agent's architecture. They illustrate their model in the scenario of multiagent issues where four robots have to lift a specific object together.

Methods combining statistical data analysis with expert domain knowledge can also help perform personalization in HRI. This is BN's case, a graphical model for data analysis and a popular representation for encoding uncertain expert knowledge in expert systems [51]. In HRI, Sekmen et al. [22] introduce a BN to learn the users' preferred beverages. In this paper, the BN's structure and the variables are incrementally learned by the network, and they essentially distinguish two processes during the learning. The first one includes the estimation of the parameters by employing the expectation maximization (EM) algorithm. The second process concerns the inference of the network, which is based on multiple variables (e.g., time, day, season, food, temperature or drink) and employs the algorithm proposed by Pearl [52]. The algorithm structures the network as a polytree where different nodes, representing the variables, are linked to each other.

Summarily, when the user interacts with the robot, the EM updates the BN nodes' conditional probabilities. On the other hand, the inference module helps the robot suggest beverages according to the BN structure and parameter values.

When the preferences of the users are relevant, various methods use these preferences as optimization parameters. Essentially, when a robot makes a decision, the user sends feedback assessing the robot's action to adjust its parameters and improve its future decisions according to the user's preferences. In the literature, this is called preferences-based optimization. For example, Roveda et al. [53] set up a pairwise preferences-based optimization in robotic sealing tasks, which tunes the robot's velocity according to the confronted geometrical features. In doing so, at each iteration, the user proposes a comparison of the global task quality between two experimental depositions (the last trial and the best one so far achieved in the optimization process). They also provide a judgment on different criteria (acceptable or not acceptable) to assess the robot's work on several aspects. In another context, Li et al. [54] employ a similar process to recommend meals in restaurants according to customers' preferences. The authors implemented a multi-attribute relation matrix tri-factorization (MARMTF) technique to recommend dishes under four criteria:

(1) the user's ordering history and their rating scores of the food on the menu, (2) the ingredients, (3) the spice level and (4) the price of the food. Although both given examples are not directly related to the domain of social robotics, these methods may be helpful when setting up personal robots that adapt their behaviors according to a user's preferences.

### 4.2. Behavior Generation

Decision-making is essential to define the actions that a robot has to take according to particular situations. However, social robots also need to adopt and employ specific social skills in order to develop appropriate behaviors accepted by users. This ability can be achieved by adapting a robot's behavior to nonverbal and verbal social cues or making the robot learn social skills. These elements are mainly based on social signals expressed by users (e.g., facial expression or speech) and permit the robot to acquire those abilities autonomously or not. Here, we depict some of those factors.

#### 4.2.1. Facial Expressions

Facial expressions are important in social interaction. They can be described as the movements of the facial muscles, which convey an individual's emotional state to observers. These nonverbal communications are a primary means of conveying social information between humans and can be employed as a social signal to adapt a robot's behavior. The methods used are mainly related to computer vision and employ state-of-the-art models that prove their performance, such as convolutional neural networks (CNN). For example, if a user feels sad, a robot could take the initiative to support them by calling a family member [25].

Facial detection is also a reliable feature for personalizing a robot's behaviors to the user's identity. For example, in Sekmen et al. [22], a robot proposes different beverages to users based on user information. The robot also reminds the user of their class and meeting times. The authors combined a NN with the eigenfaces and fisherfaces [55] approaches to recognize faces. More precisely, when a face is detected, all of the normalized face images are converted into a vector, and the means are calculated. For eigenfaces, the covariance matrix is constructed, and eigenvectors of this matrix are calculated. For fisherfaces, the within-class scatter and between-class scatter matrices are generated. Then, the generalized eigenvectors of those matrices are calculated. The eigenvectors generated by eigenfaces and fisherfaces algorithms are then fed into a three-layered NN that is trained to recognize an unknown facial image. Similarly, in [27], local binary pattern histograms (LBPH, [56]) are used for facial recognition and allow the robot to recall the name of the user during the interaction. In [29], a deep NN pretrained on thousands of images is used to extract the features from users' faces for facial recognition.

Facial expression could permit the detection of user engagement during an interaction. This is an essential signal that social robots need to consider in order to determine a user's level of interest in the conversation [31] or whether users are ready to interact with the robot [20,34]. As explained in [20], the authors depicted a list of handcrafted ML rules and supervised learning to learn engagement from an annotated human-robot corpus [37] using a set of visual sensors (two calibrated stereo cameras and a Microsoft Kinect depth sensor). A different model, such as an SVM or a multinomial logistic regression with a ridge estimator [57], could also be used. Thereupon, the authors chose the model with the best performance. Likewise, the authors in [34] used a list of methods to detect and track people in front of a robot using audio and visual cues. A module called OpenHeadPose [58] was employed to estimate the head pose. The authors fused it with a gaze-sensing module [59] to determine when users were looking toward the robot.

Facial expressions also include the detection of gaze as a parameter to adjust a robot's behavior. It may also be an indicator to increase human comprehension of a robot's behavior, as explained by Andrist et al. [24], who employed gaze aversion (when people divert away from the gaze of an interlocutor) as an element to improve conversational understanding between humans and robots. They mainly define three possible contexts

for gaze aversion: (1) when the person is thinking of responding to a question, (2) when the person is intimidated by the interlocutor or (3) to mark a pause while the user is speaking. Subsequently, they manually integrated these gazes in a humanoid robot, NAO, and employ a Kalman filter [60], a linear predictive filter used for estimating the state of a system given past states and target goals, to create smooth motions between the different gaze aversions.

### 4.2.2. Body Gestures

Body gestures are also an indicator of the human social state and can make a robot more humanlike when interacting with a person. Indeed, it was demonstrated that a robot with different gestures and voices at a different intensities affects users' subjective reactions to the robot [61]. For example, Deshmukh, Foster and Mazel [62] integrated a finer-grained method of gesture control based on sentiment, as well as a set of effectively generated artificial sounds [63], intending to enhance the expressiveness of a humanoid robot.

Similarly, in [18], the robot NAO adapted its gestures based on the user's personality. They used an external tool called BEAT (behavior expression animation toolkit) [64], a software that generates a synchronized set of gestures according to an input text, defined here by the robot's speech. It uses linguistic and contextual information in the text to control body and facial gestures, as well as the voice's intonation. BEAT is composed of different XML- based modules. The language-tagging module receives an XML-tagged text and converts it into a parse tree with different discourse annotations. The behavior-generation module uses the language module's output tags and suggests all possible gestures; then, the behavior-filtering module selects the most appropriate gestures (using the gesture's conflict and priority threshold filters). Finally, the behavior-scheduling module converts the input-XML tree into a set of synchronized speech and gestures, which is ultimately converted into some executive instructions by the script compilation, usable to animate a 3D agent or a humanoid robot.

Gestures can also be directly learned from human experiences, as defined in Yoon et al. [33]. The authors used an end-to-end NN model, including an encoder for speech-text understanding and a decoder to generate a sequence of gestures. More specifically, the encoder, a bidirectional recurrent neural network [65], captures speech context by analyzing input words one by one. The results are transmitted to the decoder to generate gesture motions. For decoding, the authors also employed a recurrent neural network with pre- and post-linear layers. Subsequently, the model is trained on the TED gesture dataset, a dataset with 1295 videos of talks from a conference. Ultimately, the posture is generated using the OpenPose methodology, which is fed to the network afterward for the training.

Similarly, in [32], the authors investigated the importance of the user's cultural background when generating social robots' emotional bodily expressions. In order to meet the requirements for cultural competence, they implemented an incremental learning model to select a representative emotional response through long-term human-robot interaction and a transformation model to convert human behavior into the Pepper robot's motion space. The proposed approach was evaluated by an experiment that lasted approximately three days. Throughout the interactions, the robot utilized user information to generate emotional behaviors that were acceptable and recognizable to a group of subjects who share the same cultural background.

### 4.2.3. Speech

Nonverbal social skills, such as facial expression and body gestures, are essential signals to estimate a human's social state and influence a robot's behavior; nevertheless, speech is just as important. Indeed, it is the primary resource of communication that humans use. It facilitates the understanding of intentions among different actors during interactions. As a result, it is essential that we focus our review on different speech models employed within the literature on social robots.

Due to the complexity of human speech, the majority of systems use human operators to control the robot's speech [11,14,17,35] or an RBS [8,10,27]. Architecture is specific to the application, although several articles employ common software to build the robot's dialogue.

Some use the Artificial Intelligence Markup Language (AIML) [8,9,22]. This is an XML dialect for creating natural language software agents, and it may be modified easily to match the role in which the robot will interact. It can include the additions of new words and phrases related to a specific sector [9] or associated with a precoded chatbot, such as ALICE (http://www.alicebot.org/, accessed on 4 October 2021). There are other precoded speech models, such as PERSONAGE, a natural language generator that adapts the generated text to the personality dimensions of the interacting human [66]. As depicted in [18], PERSONAGE is a data-driven module that takes as input a pragmatic goal and a list of real-valued-style parameters representing scores on the five personality traits. With the help of machine-learned models acquired from a dataset pairing sample utterances with human personality judgments, PERSONAGE produces an appropriate sentence according to the application domain, e.g., recommendation and/or comparison-selection of restaurants [18].

Autonomous chatbots have also demonstrated their usefulness for natural language understanding (NLU). However, their results remain more uncertain than RBS or remote control by human operators. In Foster et al. [34], a conversational interaction system is integrated into the robot Pepper to have a task-based dialogue system with chat-style open-domain social interaction to fulfill the required tasks while, at the same, time being natural. The authors used the conversational framework ALENA, combined with HERMIT NLU [67], for the NLU module to guide people in a shopping mall.

In Reig et al. [35], the Google Cloud text-to-speech (TTS) engine (https://cloud.google.com/text-to-speech?hl=en, accessed on 4 October 2021) with five different voices to generate the agents' scripted speech in advance was employed. Furthermore, a repository of Google TTS-generated common phrases were registered so that the agents could respond to unplanned deviations. Perera et al. [29] decided to use external tools for speech recognition by connecting the robot Pepper to the IBM Watson speech-to-text service (https://www.ibm.com/watson/developercloud/speech-to-text.html, accessed on 4 October 2021). This service runs on the IBM Bluemix Cloud service and requires the audio file recorded to be sent to the remote server when a person touches the robot's hand.

### 4.2.4. Interaction

More generally, researchers and roboticists could adopt a strategy to adapt robots' behaviors by influencing their interaction with the user. Indeed, the methods above have demonstrated their performance for particular situations (e.g., analyzing emotions and making decisions accordingly). However, a robot has to combine all of these factors to personalize the interaction that it will have with humans.

ASR needs to combine these methods defined above to generate accurate behaviors in a full interaction. One available solution is to use handcrafted architectures developed from scratch. However, the development time would be somewhat significant when we can use external tools to permit communication between a robot's different sensors and modules. Indeed, most papers presented in this review use the robotics operating systems (ROS), a flexible framework for writing robot software. ROS is a collection of tools, libraries and conventions that aim to simplify the task of creating complex and robust robot behavior across a wide variety of robotic platforms. ROS is well known in robotics because it provides many applications and algorithms for autonomous robots, including social robots, for perception, planning and localization. Researchers are also able to integrate their models and algorithms and evaluate them in simulations to guarantee the program's function. We can find large available libraries to control a robot's navigation and planning [25,27,29,34] by employing state-of-the-art methods for SLAM (simultaneous localization and mapping). There is also an active community in ROS in, within which

many researchers around the world provide new frameworks and algorithms that we can use for our work. In social robotics, Huang et al. [15] presented an ROS toolkit to generate social behaviors for robots. In addition to providing simulated robots with sensors, this open-source project provides social behaviors that researchers can easily import into their experiments. The authors are inspired by activity theory [68] for guiding the generation of humanlike behaviors for robots.

## 5. Discussion

In the above sections, we reviewed several papers on ASR and described how each one uses the notions of personalization, localization and adaptation. In order to support our review, we depicted a list of methods for social robots concerning decision-making and how to generate appropriate social behaviors. We also divided these terms into two different parts, since it is necessary to combine them for effective social robots. In decision-making, it was important to define algorithms from the perspective of handcrafted rules with RBS in which the model's architecture is built by rules specifically designed for the environment and the context wherein the robot will perform. On the other hand, we distinguished these models from those using new concepts in AI, such as RL or ML. Both systems present similarities, such as enabling robots to use their sensors for decision-making and effectively adapting to environments and situations. However, the expected results are not the same with AI since their outputs refer to mathematical models that use data to train and learn how to behave, such as NN or Markov decision processes with RL. Even though their results are remarkable and researchers have made promising discoveries, it is still early to be certain of their functioning, and there are still unexpected results. Conversely, RBS is a deterministic technique whereby results are controlled and expected. However, this requires ongoing work from developers and may be complex, depending on the system. By presenting both advantages and disadvantages of these methods, we hope to help researchers in their choices but also encourage them to contribute to the development of this sector.

At the same time, it is important to note that an effective social robot needs to adopt social skills that are familiar to and similar to those of humans. The list of four terms (facial expressions, body gestures, speech and interactions) is non-exhaustive but covers many possible elements that might influence a robot's behavior. We have identified some existing software and frameworks that can be used to achieve appropriate social HRI.

Finally, the different applications presented here also possess standard abilities to interact with users and respond to their needs effectively. For example, they consider if speech ability is essential to communicate with users or if the social robot can omit this ability to meet the needs of people [10]. However, most of these require speech recognition when dealing with oral communication, which significantly affects the user's interaction. Developers have set up alternative solutions to these issues, such as screen-based communication [17] or a human operator controlling the received answers when needed [14]. The type of robot you want to employ for your tasks is also an essential point to ponder before going through the software and algorithms you need to develop. For example, it is important to consider whether you desire a robot with complete mobility or whether a static robot is sufficient for your application. Consider also whether you need a humanoid robot with interactive facial expressions [21] and/or arms to perform complex gestures [34] or whether a simple mobile robot that has main functionalities to deliver objects [10] is acceptable. This consideration should include target users, as these matters are also related to users' preferences according to their culture or personality [69]. Moreover, this work enables researchers to seek additional resources to achieve personalized behaviors, such as a camera [20] to reinforce the visual recognition of a robot or sensors to have an efficient motion system [29].

## 6. Conclusions

Personalization and localization have a significant impact on how we will design and develop social robots in the future. In this paper, we have presented research that explores how adaptation could affect users who interact with them, in positive and negative ways. Across these papers, there are several methods employed in order to allow robots to achieve adaptable interactions in which they cater to the user's preferences and/or needs. We have characterized personalization by using models related to social criteria of a group of individuals (social model) or specific for each user (user models). Furthermore, we have demonstrated that these models can be either static, when the model uses only predefined knowledge about the user, or dynamic, when the model uses predefined knowledge but can update this information across interactions.

We have also shown that depending on the model employed, the algorithms and methods used are equally important. These are essential, regardless of whether they concern simple rule-based systems modeling that use predefined architecture for permitting decision-making and developing the behaviors of social robots or whether they use autonomous methods based on the current state-of-the-art techniques employed in the sector of AI (e.g., ML algorithms). This review has presented an overview of the different elements that could affect the behaviors of social robots, such as facial expression, speech or gestures. This summary and characterization of adaptive models is primarily intended to help when designing social service robots and may be useful for roboticists or researchers who are looking for methods to employ for adapting, personalizing and localizing a robot's behavior.

**Funding:** This research was funded by an Institute for Information Communications Technology Promotion (IITP) grant funded by the Korean government (MSIP) (No.2020-0-00842, Development of Cloud Robot Intelligence for Continual Adaptation to User Reactions in Real Service Environments). Funders played no role in data collection, interpretation or reporting.

**Institutional Review Board Statement:** Not applicable.

**Informed Consent Statement:** Not applicable.

**Acknowledgments:** The project was supported by an Institute for Information Communications Technology Promotion (IITP) grant funded by the Korean government (MSIP) (No.2020-0-00842, Development of Cloud Robot Intelligence for Continual Adaptation to User Reactions in Real Service Environments).

**Conflicts of Interest:** The authors declare no conflict of interest.

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
