# Peer review of "Personalization and Localization in Human-Robot Interaction: A Review of Technical Methods"

_robotics, doi:10.3390/robotics10040120_

Round 1

Reviewer 1 Report

The paper is interesting and it can be considered for publication after a major review based on the following comments:

  1. even if social robotics is the main focus of the paper, a paragraph starting the introduction can be useful to introduce the general problem, that relates to all the different domains in which humans and robots interact. Therefore, I suggest writing a small paragraph introducing the problem in general, to then focus on social robotics. Consider citing works from collaborative robotics [1], exoskeletons [2], and rehabilitation [3];
  2. even if the paper is considering social robots, there are some methodologies that have been applied in different robotics fields that can be beneficial for the considered domain. In particular, preference-based optimization can be one of them, helping to optimize the robot behavior based on the user's feedback. Consider citing [4], in which preference-based optimization is used to optimize the robot behavior based on the user's feedback;
  3. Figure 2 has a really low quality;
  4. in Section 4, why RL is outside ML?
  5. in the discussion Section, consider to put some insights on applications that have been developed;
  6. Figure 1 is not useful for the paper;
  7. check the English.

[1] Vicentini, Federico, et al. "PIROS: Cooperative, safe and reconfigurable robotic companion for CNC pallets load/unload stations." Bringing Innovative Robotic Technologies from Research Labs to Industrial End-users. Springer, Cham, 2020. 57-96.

[2] Mauri, Alessandro, et al. "Mechanical and control design of an industrial exoskeleton for advanced human empowering in heavy parts manipulation tasks." Robotics 8.3 (2019): 65.

[3] Spagnuolo, Giulio, et al. "Passive and active gravity-compensation of LIGHTarm, an exoskeleton for the upper-limb rehabilitation." 2015 IEEE International Conference on Rehabilitation Robotics (ICORR). IEEE, 2015.

[4] Roveda, Loris, et al. "Pairwise Preferences-Based Optimization of a Path-Based Velocity Planner in Robotic Sealing Tasks." IEEE Robotics and Automation Letters 6.4 (2021): 6632-6639.

Author Response

We thank reviewers for their thoughtful comments. We have addressed each comment and responded below. Changes are signified in the document, using track changes.

Author Response

(The authors gave the same response as above.)

Round 2

Reviewer 1 Report

Authors revised the paper. It can now be accepted.

Author Response

Authors revised the paper. It can now be accepted.

Thank you for the confirmation.

Reviewer 2 Report

The authors have made adequate efforts to meet the reviewers' comments. I would additionally suggest to include some words about users preferences learning during physical interaction, such as assistive scenarios for handicapped people (e.g. dressing, feeding), even if it is just to state that you focus on non-physical interactions, as it seems to be. A fine reference that you may cite in this sense is "Are Preferences Useful for Better Assistance?: A Physically Assistive Robotics User Study"
Gerard Canal, Carme Torras, and Guillem Alenyà
ACM Transactions on Human-Robot Interaction (THRI), 10(4): 1-19, 2021

A few typos still exist, in the text (line 723 "returnany" seems to resist to leave, and line 1372 "user(user models)"), and mainly in the references. For example, misspellings ("Martins, G.c., ,alo S., L.i. Santos, ,s,", "Mairesse, F.c., ois") or incomplete references (e.g.,  reference [63] has just authors, title and year, you should state other things as well: conference, pages, etc., there are many more examples). Moreover, it is a usual practice to list all the authors of a paper in the references, the "et al." is generally left for citations within the main body of the paper, but this may depend upon the particular policy of the journal.

Author Response

The authors have made adequate efforts to meet the reviewers' comments. I would additionally suggest to include some words about users preferences learning during physical interaction, such as assistive scenarios for handicapped people (e.g. dressing, feeding), even if it is just to state that you focus on non-physical interactions, as it seems to be. A fine reference that you may cite in this sense is "Are Preferences Useful for Better Assistance?: A Physically Assistive Robotics User Study" Gerard Canal, Carme Torras, and Guillem Alenyà
ACM Transactions on Human-Robot Interaction (THRI), 10(4): 1-19, 2021

We have added some lines for the preferences learning in the “User model” Section between lines 348-355 according to the paper you advised to cite.

A few typos still exist, in the text (line 723 "returnany" seems to resist to leave, and line 1372 "user(user models)"), and mainly in the references. For example, misspellings ("Martins, G.c., ,alo S., L.i. Santos, ,s,", "Mairesse, F.c., ois") or incomplete references (e.g.,  reference [63] has just authors, title and year, you should state other things as well: conference, pages, etc., there are many more examples). Moreover, it is a usual practice to list all the authors of a paper in the references, the "et al." is generally left for citations within the main body of the paper, but this may depend upon the particular policy of the journal.

Thank you. We have made sure to take into account those changes. We have also made major change for the references by correcting the bibliography. We have added the conference name in most of the papers with the location (if specified). We also added the publisher name, when it was available.
